mechanical engineering

gas hydrate, deepwater gas well, cavitation, computational fluid dynamics, experimental study

**Author for correspondence:**
Mingbo Wang
e-mail: wangmb@upc.edu.cn

# Towards the development of cavitation technology for gas hydrate prevention

## Mingbo Wang[1,2], Junjie Qiu[1,2] and Weiqing Chen[3]

[1]Key Laboratory of Unconventional Oil and Gas Development (China University of Petroleum (East China)), Ministry of Education, Qingdao 266580, People's Republic of China
[2]School of Petroleum Engineering, China University of Petroleum (East China), Qingdao 266580, People's Republic of China
[3]College of Petroleum Engineering and Geosciences, King Fahd University of Petroleum and Minerals, Dhahran 31261, Kingdom of Saudi Arabia

MW, 0000-0002-4885-9960

In offshore gas well drilling and production, methane hydrate may block the tubing, resulting in the stoppage of gas production. Conventional methods such as injection of thermal hydrate inhibitors, thermal insulating or heating, gas dehydration and reducing pressure are time-consuming and expensive, and sometimes, they are not realistic in production conditions. New methods are needed to lower the cost of gas hydrate prevention and to overcome these limitations. The thermal effect of cavitation was applied to the prevention of gas hydrate in this study. The thermal impact of cavitation, supposed to heat the fluids and prevent the formation of gas hydrate, was evaluated. Numerical simulation was performed to study the thermal performance of cavitation. Furthermore, experimental studies of the influence of initial temperature, flow rate, fluid volume and fluid viscosity on the thermal effect of cavitation were performed, and the results were analysed.

## 1. Introduction

Offshore oil and gas reserves account for more than one-third of the overall energy resources and have globally gained more attention. Most of these reserves are located in the following areas: the Gulf of Mexico, the Gulf of Guinea, the Persian Gulf, the North Sea near the UK and the South China Sea. Data from the United States Geological Survey show that the offshore oil reserves to be recovered are 54.8 billion tons, and the offshore natural gas reserves to be recovered are 78.5 trillion cubic metres (the USA not included).

Natural gas plays an essential role in human energy utilization and may become one of the leading energy sources for human development in the near future. Natural gas not only has the

characteristics of high efficiency and cleanliness but also has higher economic sustainability than coal and oil. Therefore, as the primary energy source in the future, its demand is increasing [1].

Gas hydrate formation is a primary safety issue in the oil and gas industry, especially in offshore gas production and transportation. A gas hydrate or natural gas hydrate is an ice-like, non-stoichiometric, clathrate crystalline compound composed of free water and natural gas mixed at high pressure and low temperature under appropriate temperature, pressure, gas saturation, water salinity and pH value [2]. The polyhedral cage structure formed by water as the host molecule through van der Waals force fills the guest molecules with suitable structure in the cage to maintain its thermodynamic stability [3]. In nature, it exists as a crystalline solid on the seabed or permafrost. To form natural gas hydrate, the following needs to be met: (i) the presence of free water; (ii) low temperature, the gas temperature must be equal to or lower than the dew point temperature of water; (iii) high pressure, both the temperature and pressure provide thermodynamic phase equilibrium conditions for gas hydrates; and (iv) other conditions include abrupt changes in flow rate or direction, throttling effects, pressure fluctuations, flow disturbances and the presence of sour gases such as $H_2S$ and $CO_2$.

There are many low-temperature and high-pressure locations in the development and transportation of natural gas. These locations are likely to form natural gas hydrate and block the production pipeline and valves, posing severe threats to the safe production of natural gas. The formation of hydrates during oil and gas production, especially in offshore gas wells, will cause pressure fluctuations inside the pipelines, which can cause potential flow assurance problems and safety hazards for future downhole operations. It not only affects production, but also creates unexpected operational challenges, such as shutdowns, explosions and personnel safety concerns [4]. Taking an ultra-high pressure sour gas well as an example, due to the high pressure around the wellhead and the gas content of $H_2S$ and $CO_2$ acidic components, it is easier to form natural gas hydrates around the wellhead, and the formation speed is fast, and the blockage is dense, which seriously affects the production process and even endangers the safety of gas wells. As the hydrate plug is formed and the flow is blocked, the pressure difference will arise between the lower and upper parts of the hydrate plug. Once the plug is removed, the successful removal of the hydrate will release the ultra-high pressure gas flow immediately, and the strong impact on the pipeline or valves may cause operators more significant risks and huge costs. The oil and gas companies spend more than US$200 million annually to prevent hydrate formation and aggregation [5]. A hydrate plug of several hundred metres long can form quickly. The locations and severity of natural gas hydrate deposition or plugs depend on several factors, such as production rate, geothermal profile and fluid composition.

Several strategies have been adopted to handle the gas hydrates issues [4]. Both physical and chemical methods have found their applications in hydrate prevention and treatment [6]. Physical method alters the temperature or pressure environment through specific technical means. Physical methods include dehydration, thermal insulating, heating and depressurization. The dehydration technology will dry the gas before passing it into the pipeline so that there is no water in the gas pipeline, thereby reducing the possibility of hydrate formation in the transportation pipeline. However, it is not suitable for the production process of gas wells because of the complicated downhole dehydration equipment and limited reliability. Thermal insulating is to wrap an insulation layer on the outside of the pipeline to isolate the pipeline from the ambient environment and to maintain the internal temperature of the pipeline without reaching the equilibrium temperature of the hydrate. It is suitable for short-distance offshore pipelines. Insulated tubing has found limited applications in offshore natural gas wells or pipelines because of its high cost and short service life. Heating and depressurization are mainly aimed at the situation where hydrate has formed in the wellbore or pipeline. By heating or depressurizing the blocked section, the thermodynamic conditions in the pipeline are changed to promote the decomposition of the hydrate. For both pipeline transportation and downhole production, depressurization usually means reduced throughput or production rate.

Chemical methods achieve inhibition by altering the phase equilibrium conditions for hydrate formation or by slowing down the rate of hydrate formation and conversion, generally by adding inhibitors to the system. Inhibitors can be divided into thermodynamic inhibitors (THIs), and low-dose inhibitors (LDIs), and LDIs can then be subdivided into kinetic inhibitors (KIs) and anti-aggregation (AA) agents [7,8]. THIs prevent the formation of hydrates by altering the phase equilibrium conditions for hydrate formation and are commonly used, such as alcohols (methanol, ethylene glycol, diethylene glycol, etc.) and salts (sodium chloride, potassium chloride, calcium chloride). The addition of THIs (e.g. methanol, ethylene glycol, etc.) to the system can break the phase equilibrium of hydrates at specific temperature and pressure, causing hydrate formation to move to higher pressures or lower temperatures, thus eliminating the possibility of hydrate formation in the

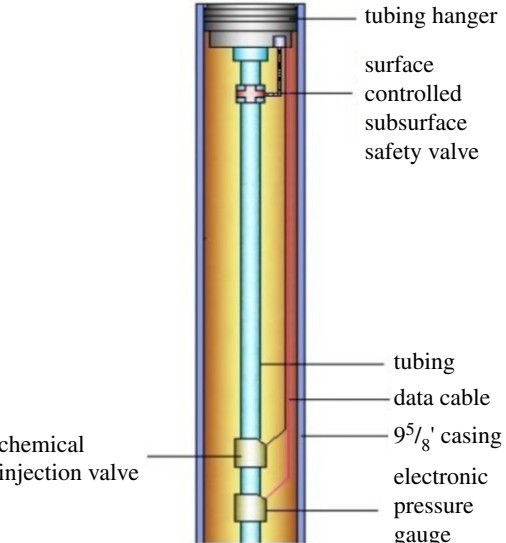

**Figure 1.** Tubing configuration for the chemical injection method.

wellbore or pipeline [9,10]. However, the injection volume of THIs is large (10–50 wt% of the water content in the system), with high requirements for storage, injection and recovery equipment during the production process, complex production processes, high production costs, specific toxicity and poor environmental friendliness.

KIs based on water-soluble or water-dispersible polymers can slow down the nucleation time or hinder the growth of hydrates, thereby effectively inhibiting the nucleation growth rate of hydrates or preventing their further aggregation. Kinetic inhibition represented by polyvinylpyrrolidone (PVP) was first applied in tetrahydrofuran hydrate inhibition, where it could slow down the growth and aggregation time of tetrahydrofuran hydrate [11,12]. Subsequently, the researchers synthesized a series of polymers using PVP as the mainstay, many of which have better effects than PVP, such as polyvinylcaprolactam (PVCap) [13–17]. For the time being, KI is available in the form of ethylene caprolactam, amide polymers, hyperbranched polyamides, chlorinated polymers, ionic liquids and other types [10,18]. In addition, some natural products have been found to have kinetic inhibitory effects, such as biological antifreeze, tapioca starch, fructose, amino acids, etc. These are emerging as KIs with environmentally friendly characteristics and are, therefore, also known as green KIs. The dosage of KI is very low, about 0.1–5 wt% of the water content of the system; the overall cost is about 50% lower than that of THIs. The macroscopic inhibitory effect of kinetic inhibitors has been verified. Inhibition failure occurs at high subcooling levels (greater than 10 K), and biodegradability is not good enough for kinetic inhibitors.

AA agents are some polymers and surfactants based on non-ionic amphiphilic compounds [19]. By preventing the aggregation of hydrate crystal particles, they can enhance flow stability in the pipeline, reduce the contact and deposition among hydrate particles, thereby delay the aggregation of hydrate in the pipeline and maintain the flow condition of the pipeline. AA agents are generally used under oil–water coexistence conditions. Typical AA agents include diethanolamine, ethoxylated amine surfactants, quaternary ammonium salt surfactants, caprolactam surfactants and alkyl amide surfactants [20–22]. The dosage of AA agents ranges from 0.5 to 2 wt%, which is much lower than THIs, but they are expensive and have a limited effect on the dispersion of slurry particles. A previous study has revealed that AA agents may be invalid when water cut is more than 50 vol%. In practice, they are usually used in combination with other inhibitors [9].

For the inhibitors to take effect, commercial software packages (such as PIPESIM and OLGA) can first be used to determine the most probable locations of gas hydrates. Inhibitors are then injected continuously into the mainstream to mix with the natural gas (figure 1).

Hydrate crystals trap and temporarily store large quantities of methane gas molecules, and when the thermodynamic conditions change, the hydrate crystals then are decomposed, and these gas molecules are released, bringing us considerable natural gas energy. Many of the above hydrate control methods can be used directly without modification for the efficient development of hydrate resources, such as depressurization and thermal injections. The USA, Canada, Germany, Russia, Japan, India and other

countries have all carried out hydrate-related development work. Specifically, the USA has conducted hydrate test trials in Alaska, Japan in Nankai and China in the Shenhu of the South China Sea with various development achievements [1].

Because there are limitations to applying existing hydrate prevention and development methods, new methods are required.

Cavitation is the phenomenon of a vapour bubble formed in a liquid when the local pressure within the fluid is lowered to its saturated vapour pressure under specific conditions. Cavitation itself goes through several phases: microbubble incipiency, bubble development, bubble collapse and micro-jet [23]. When the cavity bubbles collapse, extremely high temperature and high pressure are generated inside the fluid, and the ambient liquid is then heated. For the gas hydrate to form, high pressure and low temperature are two indispensable conditions. If only the temperature remains above its critical value, the gas hydrate will not develop, or it only forms with difficulty. In this work, a cavitation method for gas hydrate prevention is proposed. The thermal effect of cavitation was studied numerically and experimentally.

There are several ways of generating cavitation: acoustic cavitation (AC), laser-induced cavitation, hydrodynamic cavitation (HC) and others. Among all these methods, AC and HC are preferred. AC involves the growth mechanism of tiny bubbles in an ultrasonic liquid field [24]. A transducer is used to transform the electrical energy to sonic energy and, finally, the cavitation energy. In AC, when the frequency is relatively low, the chemical effects prevail in the cavitating process, and radicals are produced; in return, the reaction is accelerated.

On the contrary, when the frequency is relatively high, the physical effects surpass the chemical impacts, and tiny emulsion droplets are visible between an immiscible interface that lowers or eliminates the resistance of mass transfer [25]. The tensile force or shear effect caused by physical effects can reduce the fluid viscosity [26,27]. AC has a relatively wide application among industries, with the disadvantage of a high power requirement. Typical studies of AC include research on cleaning, flux improvement of ultrafiltration processes, wastewater treatment, heavy oil and bitumen pretreatment, mechanoluminescence and sonoluminescence and desulfurization of heavy oil, etc.

To form a deep understanding of the cumulative properties of multi-bubble systems, Ashokkumar [24] performed an overview and discussed the characteristics of AC bubbles experimentally. Yusof *et al.* studied the cleaning application with ultrasonic cavitation and found that the physical effects may find application in the ultrafiltration process. And for the deactivation of pathogens, several effects work together to provide the best performance [28]. Tamura & Hatakeyama discussed the ultrasonic pressurization mechanism and found that the radiation pressure and harmonic frequency are generated around the bubble surface. Periodical pressure fluctuations on liquid pressure were observed during the experiment [29]. Badmus *et al.* analysed the existence of persistent organic pollutants and studied their effects on human beings, and they discussed the advantages and disadvantages of several treatment techniques. Review work has also been done on treating the persistent organic pollutants in wastewater using HC technology, and the direction for future research was highlighted [30]. Mohapatra & Kirpalani investigated the influence of sonication frequencies on AC performance, and they also studied the performance of AC on asphaltene content, rheological variation and metal percentage of asphalt. They found that a notable decrease in asphaltene content can be detected; liquid viscosity and shear stress are also reduced. The sonication treatment of asphalt can decrease the H/C ratio under different power inputs and frequencies. Characterization of asphaltene revealed that reduced or retarded asphaltene formation can lower the metal content in bitumen [26]. Flannigan & Suslick [31] discovered the sonoluminescence of a single bubble in sulfuric acid, and plasma was detected during the cavitating process. The fuel desulfurization process with ultrasonic treatment was detailed, and different factors on reaction mechanism and kinetics were discussed [32]. Choi explained the fundamentals of different scenarios about the sonic field and cavitating bubbles. Experimental observations have demonstrated that there exists a close relationship between the bubble dynamics and the sonoluminescence intensity. Sonoluminescence studies on the atom emissions suggested that the side of alkali-metal atom emission is closely related to the gas in the bubbles [33].

Laser-induced cavitation can be found when a pulsed beam of a ruby laser is focused on a liquid, such as water. These bubbles are created by forcing the dielectric breakdown of the fluid into plasma [34]. Bubble motion induced by a laser was studied numerically by Akhatov *et al*. With the help of the model they proposed, the collapsing and rebounding characteristics of different bubble diameters were investigated and compared with experimental data available; good agreement was observed [35,36]. Quinto-Su *et al.* generated a two-dimensional spatial cavitating bubble with the help of a light modulator. The cavitation bubbles were generated over $380 \times 380\ \mu m^2$ in an ink solution.

The proposed method could position a bubble of up to 34 µm using laser energy of 56 µJ [37]. Song *et al*. studied the cleaning effect of laser-induced cavitation. The particle removal was dominated by both the shockwave and the micro-jet generated around the laser focusing point during the collapsing process of a bubble. The increase in laser fluence has a positive effect on the cleaning efficiency, and the increase in standoff distance (the distance between the target plate and laser focusing point) decreases the cleaning efficiency [38].

HC has become increasingly popular. It involves the vaporization process of liquid and bubble development under the condition of a flowing liquid when a sudden local pressure fluctuation takes place [39]. HC can be generated with either cavitating nozzles or rotating propellers. In this paper, the rotating propeller of HC was used to evaluate the thermal properties of the HC method. Arrojo & Benito studied HC theoretically, trying to identify the dominant parameters and analyse their effects on the cavitating process. Time scales attribute primarily to the cavitating process. Compression and/or rarefaction can generate several opposing results different from what researchers have found in AC or ultrasonic cavitation [40]. The use of AC in the field of water treatment has been widely deployed, while HC as a sole technology is still immature and not very popular. Dular *et al*. applied HC to the water treatment to remove pharmaceuticals and viruses from water and wastewater. Different types of HC are required to handle or remove various pollutants [41]. Naseri *et al*. performed simulation studies on the multiphase flow inside different nozzle configurations, emphasizing how the viscoelasticity influenced the whole flow regime. In the case of step nozzle, during the incipient cavitation period, small-scale flow structures, fluctuations in flow rate and shedding frequency during the incipient are all suppressed by the fluid viscoelasticity. In the case of an injector nozzle, the volume fraction of the vapour bubble or cavity has been reduced by the viscoelasticity effect. The influence of viscoelasticity on the cavitating process is closely related to whether the cavitating vortices are aligned with the mainstream flow direction [42]. Capocelli *et al*. performed both theoretical and experimental studies on the chemical effects of HC in a Venturi-type reactor. With the help of the model proposed, the production of hydroxyl radicals can be simulated and calculated. The degradation of *p*-nitrophenol was conducted experimentally in a Venturi reactor at different inlet pressures. The optimal configuration from numerical simulations was validated experimentally [43]. HC phenomena inside a Venturi tube were investigated by Soyama and Hoshino with an emphasis on the chemical effect of HC. The relationship between the turbulent cavitating intensity and the cavity collapsing pressure downstream was investigated, and the relationships between the power input, luminescence and cavitation number were found [44]. An orifice-based hydrodynamic cavitator was used to study the disintegration process of waste-activated sludge. The results showed that after 150 min of cavitation, disintegration degrees of 32–60% were obtained, and the optimal cavitation number and orifice diameter selected for the disintegration of waste-activated sludge were 0.2 and 3 mm [45]. Ferrari reviewed AC and HC phenomena. The available expressions of sound speed in AC were discussed. Different discharge coefficient formulae in hydrodynamic cavitation were analysed [46].

Tritium decay activity in the cavitating process was noted and detected by Taleyarkhan *et al*. The neutron emission near 2.5 million electronvolts was also observed in the experiments. The hydrodynamic shock code simulations indicated highly compressed, hot cavitation bubble collapsing conditions suitable for nuclear fusion process [47].

To prevent or eliminate the natural gas hydrate blockage in a new way, the authors developed a conceptual design of tubing configuration, as shown in figure 2. Propeller HC equipment (cavitation heater in figure 2), working as the power source, is located around the wellhead to heat the liquid. Then, the heated fluid (water or oil) flows inside an umbilical pipe reeled around the tubing to deliver the heat needed to keep the temperature and pressure conditions away from the possible block area. The whole system is a closed loop, and the temperature can be modulated at the surface by adjusting the power input of the electrical motor. Therefore, the current study performed here emphasizes how to determine, by a computational fluid dynamics (CFD) method, the feasibility of the thermal effect around a propeller. Further experiments were performed to investigate how the operating parameters, such as initial temperature, flow rate, fluid volume and fluid properties, influence the thermal yield.

## 2. CFD simulation of cavitation around a propeller

CFD simulation has found wide applications in architectural engineering [48], hydrogen production [49], engine performance [50] and heat exchanger optimization [51]. Models and schemes have been

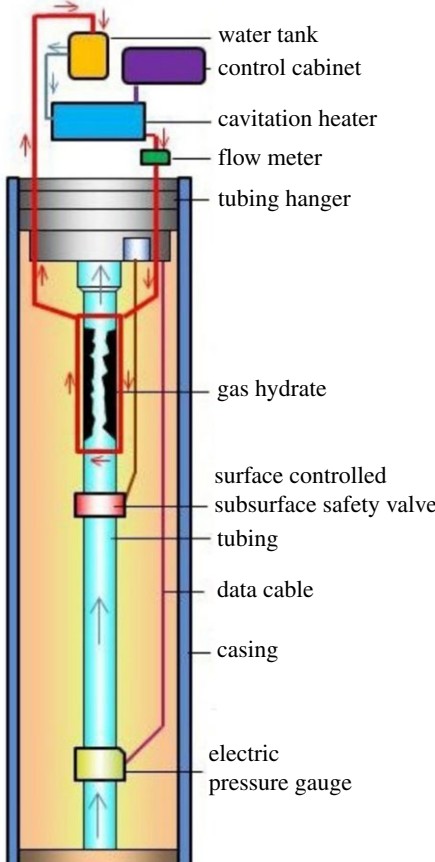

**Figure 2.** Tubing configuration for the cavitation method.

thoroughly evaluated, the calculation speed and accuracy are significantly improved. With the help of the CFD method, the fluid flow and cavitation development inside the cavitating generator is calculated, and the results are then analysed.

## 2.1. Propeller parameters

HC method was applied to generate the cavitation, and figure 3 shows the dimensions of the propeller to be modelled in the simulation. The details of the propeller are shown in table 1.

## 2.2. Mathematical model

### 2.2.1. Control equations for the gas–liquid mixture

As the local pressure of fluid decreases to its saturated vapour pressure, cavitation bubbles will form inside the fluid and the whole flow regime has changed from a single phase to multiple phases. The mixture model was chosen to address the complexity of interphase reaction and phase change phenomena.

The continuity equation for the gas–liquid mixture (subscript $m$ stands for mixture) is

$$\frac{\partial \rho_m}{\partial t} + \nabla \cdot (\rho_m \boldsymbol{v}_m) = 0, \tag{2.1}$$

where $\boldsymbol{v}_m$ is the mass-averaged velocity of the mixture.

$$\boldsymbol{v}_m = \frac{\sum_{k=1}^{n} \alpha_k \rho_k \boldsymbol{v}_k}{\rho_m}. \tag{2.2}$$

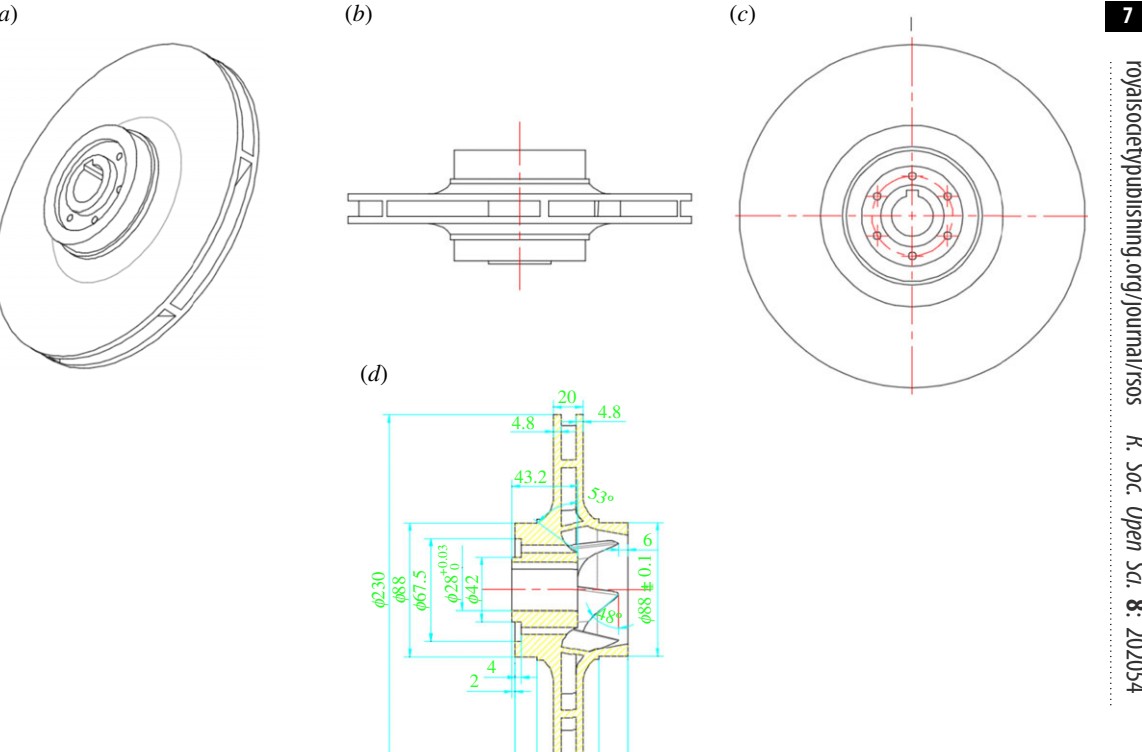

**Figure 3.** Dimensions of the propeller.

**Table 1.** Propeller parameters.

| parameter | value |
| --- | --- |
| inlet diameter (mm) | 67.5 |
| inlet angle (°) | 11.2 |
| outlet angle (°) | 36 |
| outlet blade width (mm) | 10.4 |
| number of blades | 6 |
| outlet diameter (mm) | 230 |

The mixture density can be calculated as

$$\rho_m = \sum_{k=1}^{n} \alpha_k \rho_k,$$

(2.3)

where $\alpha_k$ is the volume fraction of specific phase $k$.

The momentum equation for the gas–liquid mixture is

$$\frac{\partial}{\partial t}(\rho_m \boldsymbol{v}_m) + \nabla \cdot (\rho_m \boldsymbol{v}_m \boldsymbol{v}_m) = -\nabla p + \nabla \cdot [\mu_m(\nabla \boldsymbol{v}_m + \nabla \boldsymbol{v}_m^{\mathrm{T}})]$$

$$+ \rho_m \boldsymbol{g} + \boldsymbol{F} + \nabla \cdot \left( \sum_{k=1}^{n} \alpha_k \rho_k \boldsymbol{v}_{drk} \boldsymbol{v}_{drk} \right),$$

(2.4)

where $n$ denotes the total number of interacting phases, $\boldsymbol{F}$ stands for the body force and $\mu_m$ is the

mixture viscosity

$$\mu_m = \sum_{k=1}^{n} \alpha_k \mu_k, \tag{2.5}$$

$\boldsymbol{v}_{\mathrm{dr}k}$ is the drift velocity for specific phase $k$.

$$\boldsymbol{v}_{\mathrm{dr}k} = \boldsymbol{v}_k - \boldsymbol{v}_m. \tag{2.6}$$

The volume fraction expression for the secondary phase $p$ is

$$\frac{\partial}{\partial t}(\alpha_p \rho_p) + \nabla \cdot (\alpha_p \rho_p \boldsymbol{v}_m) = -\nabla \cdot (\alpha_p \rho_p \boldsymbol{v}_m dr_p). \tag{2.7}$$

## 2.2.2. Turbulence treatment

The shear stress transport (SST) $k$–$\omega$ turbulence model was chosen to modulate turbulent interactions between liquid and gas phases. It falls into the group of eddy-viscosity models. Different from the conventional $k$–$\varepsilon$ model which relies heavily on the application of a wall function to handle the boundary layer, the specialized $k$–$\omega$ expressions can be applied directly down to the near-wall region, and it can switch to a standard $k$–$\varepsilon$ description in the mainstream and thereby bypasses the drawback of being too sensitive to the inlet boundary conditions. The SST $k$–$\omega$ turbulence model shows superior performance over flows with fluid separations and large pressure gradients [52]. For the standard $k$–$\varepsilon$ models, a previous study has revealed that it is difficult or impossible to capture the cavitation bubble detachment due to the fact that the high viscosity predicted from the model may dampen or retard the cavitation instabilities [53].

The function details used in the SST $k$–$\omega$ turbulence model are as follows

$$F_1 = \tanh(\mathrm{arg}_1^4), \tag{2.8}$$

$$\mathrm{arg}_1 = \min\left[\max\left(\frac{\sqrt{k}}{\beta'\omega y}, \frac{500\nu}{y^2\omega}\right), \frac{4\rho k}{CD_{k\omega}\sigma_{\omega 2}y^2}\right] \tag{2.9}$$

$$\text{and} \quad CD_{k\omega} = \max\left(2\rho\frac{1}{\sigma_{\omega 2}\omega}\frac{\partial k}{\partial x_j}\frac{\partial \omega}{\partial x_j}, 10^{-10}\right). \tag{2.10}$$

The control equations for $v_t$, $k$ and $\omega$ are

$$v_t = \frac{a_1 k}{\max(a_1\omega, SF_2)}, \tag{2.11}$$

$$\frac{\partial \omega}{\partial t} + u_j\frac{\partial k}{\partial x_j} = \frac{\partial}{\partial x_j}\left[(\nu + \sigma_k v_t)\frac{\partial k}{\partial x_j}\right] + G_k - \beta'k\omega, \tag{2.12}$$

$$F_2 = \tanh(\mathrm{arg}_2^2) \tag{2.13}$$

$$\text{and} \quad \mathrm{arg}_2 = \max\left(\frac{2\sqrt{k}}{\beta'\omega y}, \frac{500\nu}{y^2\omega}\right), \tag{2.14}$$

where $F_1$ and $F_2$ are blending functions, $v_t$ the turbulent viscosity, $k$ turbulence kinetic energy, $S$ strain rate magnitude and $\omega$ the specific dissipation. The model constants used in the current study are: $\beta' = 0.09$, $a_1 = 5/9$ and $\sigma_{\omega 2} = 0.856$.

The subscript 1 denotes the $k$–$\omega$ model in the near-wall region, while 2 stands for the $k$–$\varepsilon$ description in the fully developed mainstream area.

## 2.2.3. Cavitation model

The Zwart–Gerber–Belamri model was chosen to handle the cavitation bubble dynamics with assumptions that (i) the bubble diameter is constant, (ii) the surface tension on the bubble can be neglected, and (iii) the mass transfer rate $R$ is related to the bubble number per unit volume [54]

$$R = F\frac{3\alpha_v\rho_v}{R_B}\sqrt{\frac{2|P_v - P|}{3\rho_l}}\mathrm{sign}(P_v - P), \tag{2.15}$$

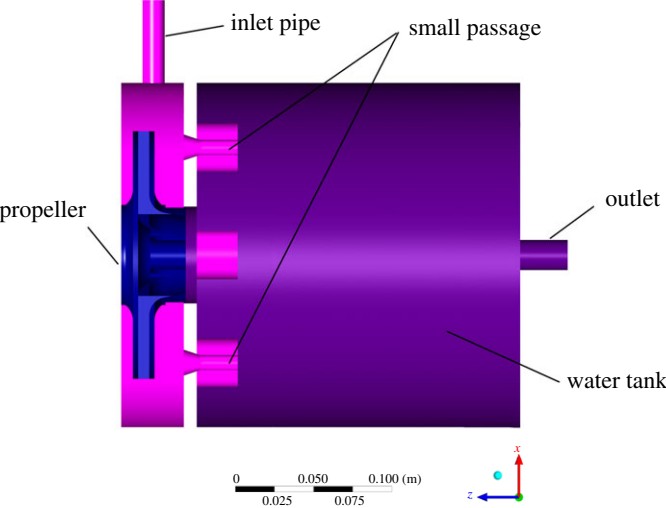

**Figure 4.** Physical model.

where $F$ is the constant coefficient, $\alpha_v$ is the volume fraction for gas phase, $\rho_v$ is the density of gas phase, $P_v$ is the pressure inside a cavity, $P$ is the surrounding ambient pressure, $\rho_l$ is the fluid density and $R_B$ is the cavity bubble diameter, $R_B = 10^{-6}$ m.

Correlation is done by substituting $\alpha_{\text{nuc}}(1 - \alpha_v)$ for $\alpha_v$, and the control equations for the phase change rate are as follows:

$$R_e = F_{\text{vap}} \frac{3\alpha_{\text{nuc}}(1 - \alpha_v)\rho_v}{R_B} \sqrt{\frac{2(P_v - P)}{3\rho_l}}(P \leq P_v) \tag{2.16}$$

and

$$R_c = F_{\text{cond}} \frac{3\alpha_v\rho_v}{R_B} \sqrt{\frac{2(P - P_v)}{3\rho_l}}(P > P_v), \tag{2.17}$$

where $\alpha_{\text{nuc}}$ is the volume fraction of gas nuclei, $\alpha_{\text{nuc}} = 5 \times 10^{-4}$, $F_{\text{vap}} = 50$ is the vaporization coefficient, $F_{\text{cond}} = 0.01$ is the condensation coefficient, $R_e$ and $R_c$ are phase change rates for bubble generation and collapse, respectively.

## 2.3. Physical model and boundary conditions

Figure 4 shows the physical model. The fluid is sucked into the heating zone equipped with a propeller through the upper-left inlet pipe with an internal diameter of 21.25 mm. The liquid then enters the passages formed by the blades, the tiny holes connecting the heating zone with the water tank, and, finally, the water tank on the right. A pipe exit is located parallel to the axis of the water tank with the same internal diameter as the inlet pipe.

The parameter details used in the steady simulation are shown in table 2.

## 2.4. Results analysis

Hexahedron meshes were used to discretize the flow domain into small cells because of their reasonable memory usage and accuracy level, as well as a relatively high mesh quality. A mesh independence check was performed with mesh density varied from 3.35 million to 6.90 million, and figure 5 shows the averaged gas fraction with different mesh densities. With the increase in mesh density, the averaged gas volume fraction tends to stabilize. A further rise in mesh density is not recommended for relatively high memory usage and a much longer calculation period. A mesh density of 4.76 million was finally chosen for the following simulation.

Static pressure contours on both the central plane and near the propeller are shown in figure 6. The coordinate system in the lower right corner offers the angle of the observation, and the coordinate system follows the right-hand rule. The propeller takes a counterclockwise rotation, as shown in the right part of

**Table 2.** Simulation parameter details.

| parameters | value |
| --- | --- |
| saturated water vapour pressure (Pa) | 2334.6 |
| water temperature (°C) | 20 |
| water density (kg m$^{-3}$) | 998.23 |
| rotational speed (r.p.m.) | 2930 |
| inlet pressure (kPa) | 108 |
| outlet velocity (m s$^{-1}$) | 0.8 |
| pressure–velocity coupling | SIMPLE algorithm [55] |

figure 6. The highest pressure lies around the propeller, while the lowest pressure is inside the water tank. The pressure around the propeller increases along the radial direction. As the propeller rotates, it pushes the fluid away from the centre, which in turn causes a pressure increase around the propeller edges. As in the working mechanism of a centrifugal pump, fluid is sucked, enters the passages among blades from the left and is forced out of the passages in the radial direction (the suction port is located on the right side of the rightmost part of figure 3). The pressure across the impeller drops and then recovers as the fluid travels away from the impeller. Compared with the suction section, the eye of the impeller is smaller; as the pump impeller is rotating, the liquid pressure is increasing. Next, the pressurized fluid enters the water tank through small passages. Moreover, dense contour lines around the edge of propeller blades indicate intense turbulent mixing among fluid layers, which facilitates cavitation. Cavitation bubbles are then generated around the centre of the propeller, where a relatively low pressure exists.

The velocity distributions on the central plane and around the propeller are shown in figure 7. As the propeller rotates, it pushes the fluid away from the centre. Hot spots can then be observed near the propeller blades. After the liquid has gone through the propeller, it is forced to pass the passage like a water jet penetrating the water tank. For the fluid between two neighbouring blades, the maximum velocity lies somewhere near the periphery of the blades for the combining actions of different mechanisms. The fluid may move in the following ways. First, there is rotational motion. The larger the radius of rotation, the higher the line speed is. Second, there is a radial movement. The centrifugal force tends to push the fluid away from the centre, generating a radial fluid flow component.

Rapid rotational movement causes a rapid pressure change. As the local pressure is lowered to the saturated vapour pressure of the fluid, tiny vapour cavities or vapour bubbles are then visible. The bubble volume fractions along the central plane and near the propeller are shown in figure 8. Near the centre or the suction of the impeller, relatively low pressure witnesses the formation of cavitating bubbles. As the liquid–bubble mixture moves from the suction side towards the periphery of the propeller or the delivery side, the pressure builds up, and the cavities or bubbles collapse and generate intense shockwaves inside the fluid. Such an observation is in agreement with the cavitating flow in a centrifugal pump [53]. The liquid is then heated during the collapse of these bubbles.

The numerical simulation has demonstrated from the theoretical point of view that cavitation bubbles can form during the propeller's rotation, and heat is generated during the whole process. Further experiments on the thermal effect of the cavitation were conducted.

# 3. Experimental measurement

## 3.1. Experimental set-up

Figure 9 shows a schematic figure of the experimental facilities. An electrical motor with a control cabinet was used to modulate the power output. As the electrical motor rotates, the propeller inside the cavitation generator rotates to generate cavitation bubbles and heat the fluid. The heated fluid is then transported to the water tank for further circulation.

Experiments were conducted using the experimental set-up to analyse the effects of initial temperature, flow rate, fluid volume and fluid viscosity on the thermal effect of cavitation. Table 3 shows the parameters used in the experiments.

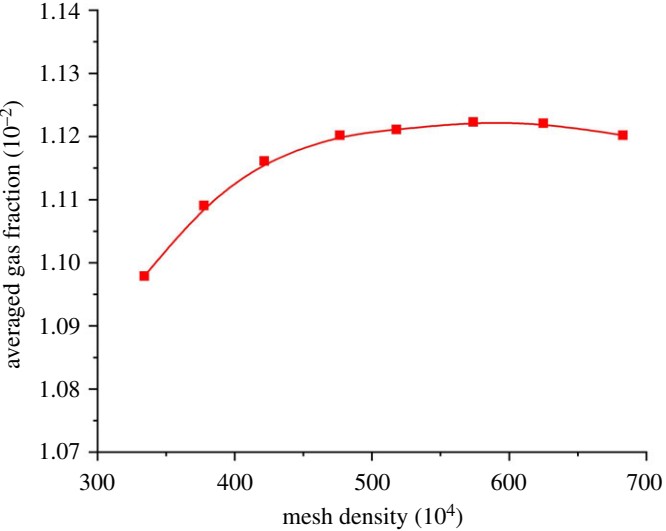

**Figure 5.** Mesh independence check.

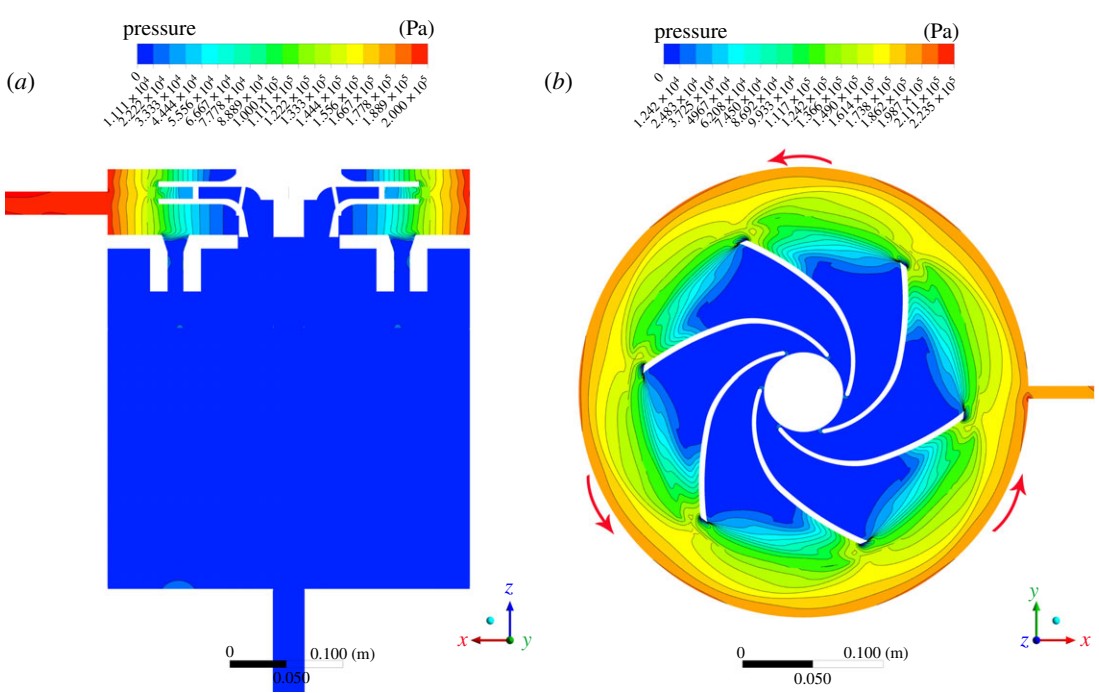

**Figure 6.** Static pressure distributions on the central plane and near the propeller (the right figure is viewed from the positive z-direction).

## 3.2. Results and discussion

### 3.2.1. Effect of initial temperature

Figure 10 shows how the fluid temperature increment develops with the increase in initial temperature. No notable change is observed on temperature increment and heat absorbed as the initial temperature increases. An increase in fluid temperature raises the saturated vapour pressure of the heated fluid, which can then facilitate the cavitating bubble generation. Meanwhile, the bubble pressure increases, and the dampening effect of bubble collapse is enhanced. These effects cancel each other out; therefore, there is little influence of initial temperature on the cavitation effect under current working conditions.

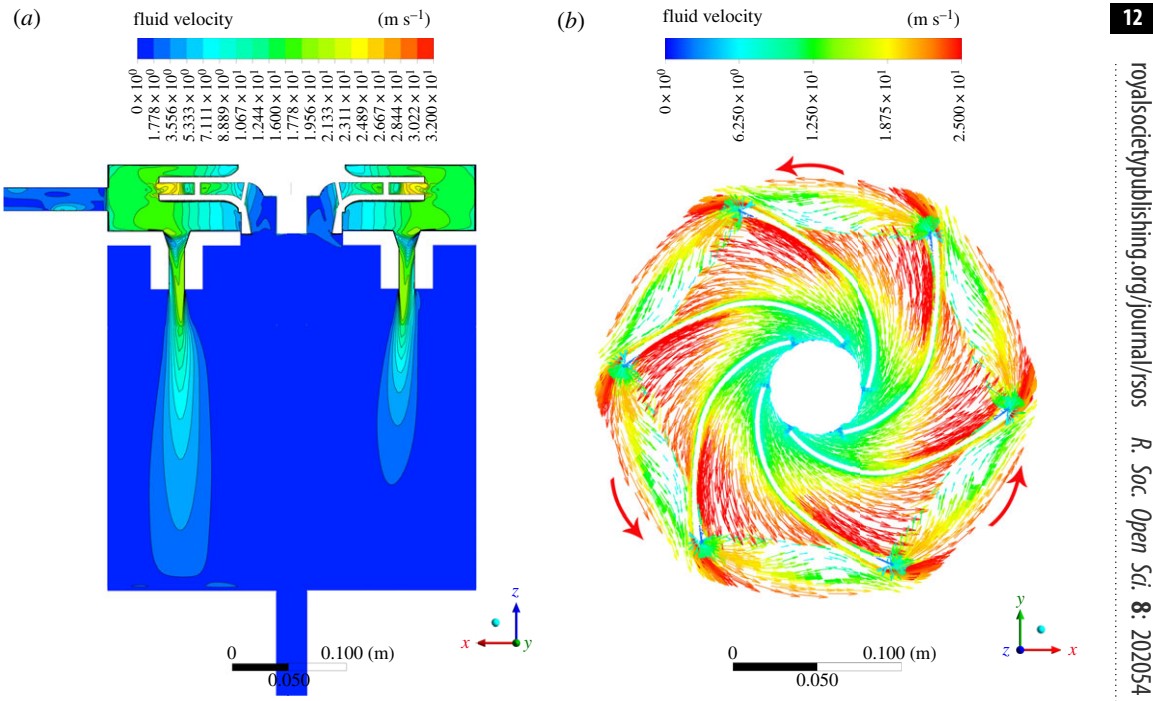

**Figure 7.** Velocity distributions on the central plane and near the propeller (the right figure is viewed from the positive z-direction).

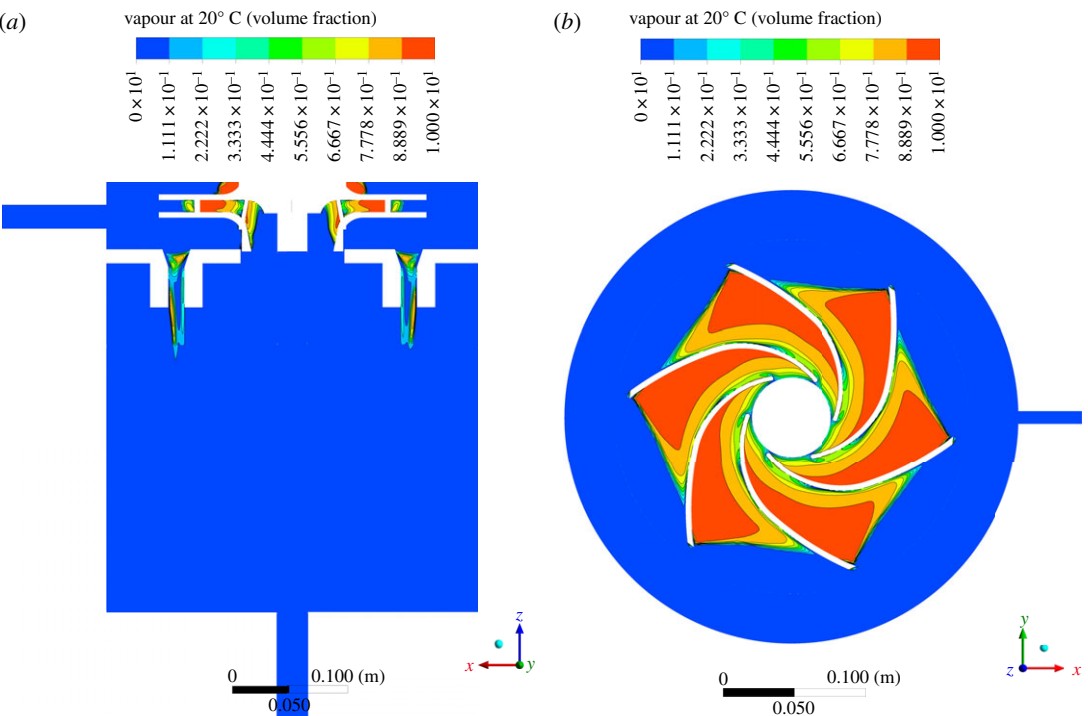

**Figure 8.** Bubble volume fraction on the central plane and near the propeller (the right figure is viewed from the positive z-direction).

### 3.2.2. Effect of flow rate

Figure 11 shows the impact of the flow rate on the thermal effect of cavitation. When the flow rate is lower than $1.6\,\mathrm{m^3\,h^{-1}}$, a notable temperature increase can be observed as the flow rate increases. As the electrical motor rotates, a cavitating area is formed around the propeller axis with its own processing

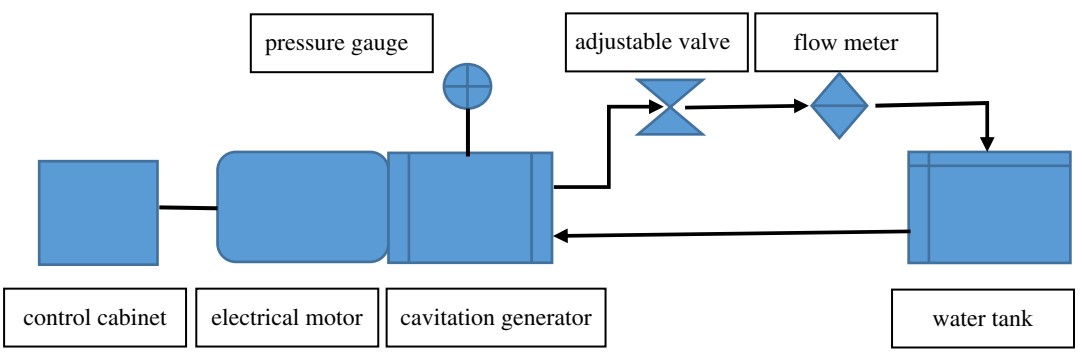

**Figure 9.** Schematics of the experimental set-up.

**Table 3.** Experimental parameters.

| influencing factors | fluid volume (l) | initial temperature (°C) | flow rate (m³ h⁻¹) | time (min) |
|---|---|---|---|---|
| initial temperature | 100 | 20 | 0.6, 1.6, 3.0 | 10 |
| | 100 | 25 | 0.6, 1.6, 3.0 | 10 |
| | 100 | 30 | 0.6, 1.6, 3.0 | 10 |
| | 100 | 35 | 0.6, 1.6, 3.0 | 10 |
| | 100 | 40 | 0.6, 1.6, 3.0 | 10 |
| flow rate | 100 | 20 | 0.6–3.0 | 10 |
| fluid volume | 80 | 20 | 1.6 | 10 |
| | 90 | 20 | 1.6 | 10 |
| | 100 | 20 | 1.6 | 10 |
| | 110 | 20 | 1.6 | 10 |
| | 120 | 20 | 1.6 | 10 |
| | 130 | 20 | 1.6 | 10 |
| xanthan gum viscosity | 100 | 20–70 | 1.6 | / |
| | 100 | 20–70 | 1.6 | / |
| | 100 | 20–70 | 1.6 | / |
| | 100 | 20–70 | 1.6 | / |
| | 100 | 20–70 | 1.6 | / |

capacity or critical flow rate (CFR). When the flow rate is lower than the CFR, an increase in flow rate usually causes a robust thermal effect of cavitation, and local hot spots are generated in the fluid and are transported downstream. While a further rise in the flow rate cannot enhance the thermal effect, a strong mixing effect happens between the heated fluid and the unheated one, and the temperature increment is reduced.

### 3.2.3. Impact of fluid volume

The impact of fluid volume on the temperature increment is shown in figure 12. Under constant flow rate and initial temperature, an increase in fluid volume causes a decrease in the temperature increment. The heat absorbed remains almost constant (figure 13).

### 3.2.4. Effect of fluid viscosity

Xanthan gum was added to the water to adjust its viscosity, and the time spent to heat the fluid to 70°C was recorded. The influence of fluid viscosity on the thermal effect of cavitation is shown in figure 14. Much more time is needed to heat the solution as the xanthan gum concentration increases. The more

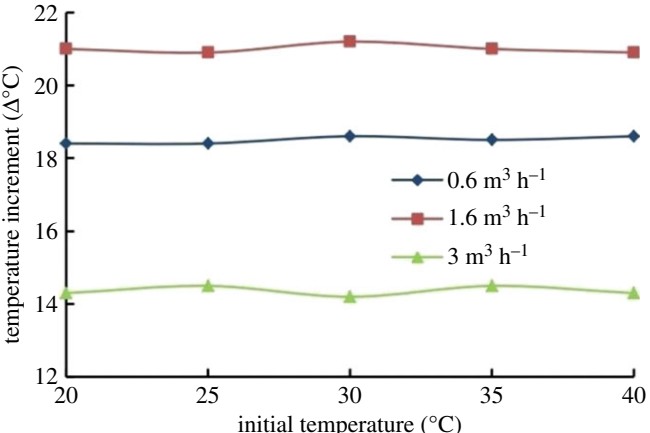

**Figure 10.** Influence of initial temperature on temperature increment.

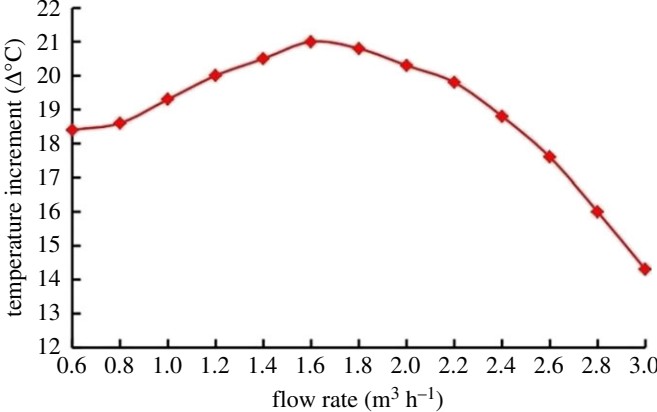

**Figure 11.** Influence of flow rate on temperature increment.

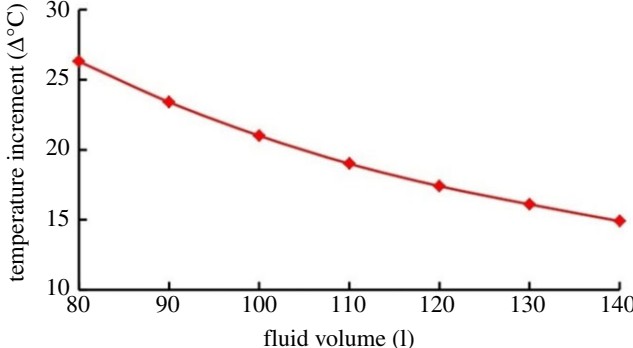

**Figure 12.** Effect of fluid volume on the thermal impact of cavitation.

xanthan gum in the solution, the higher the solution viscosity will be. As the measure of liquid resistance to deformation, an increase in viscosity usually means a higher resistance force against bubble generation, which in turn means a higher probability of suppressing cavitation, which explains why it takes a longer time to heat the solution. A bubble collapse in a fluid of high viscosity usually poses much more heat than that in a liquid of normal viscosity.

## 4. Conclusion

Conventional natural gas hydrate prevention methods are expensive and time-consuming and can even cause possible failure or adverse effects on wellbore integrity. The thermal effect of cavitation was

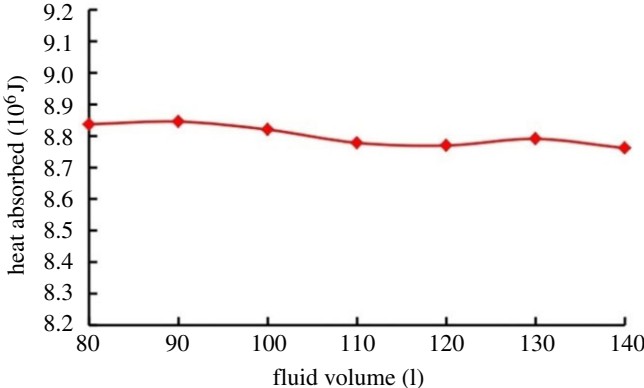

**Figure 13.** Heat absorbed under different fluid volumes.

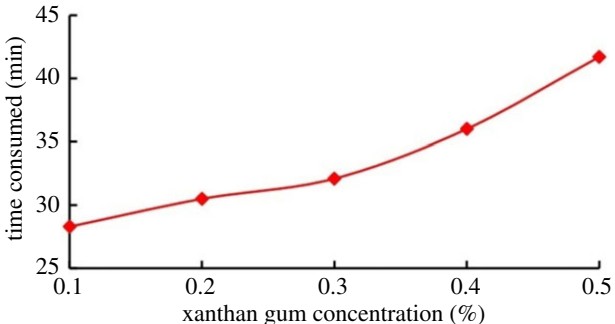

**Figure 14.** Effect of xanthan gum concentration on the time consumed to heat the solution to 70℃.

applied to the prevention of gas hydrate, and a novel method was proposed. The flow field and bubble dynamics around a propeller were simulated numerically, and further experimental investigation was performed to analyse the effect of initial temperature, flow rate, fluid volume and fluid viscosity on the thermal effect of cavitation. Cavitation bubbles were generated near the propeller centre and then transported down through the passages and, finally, into the water tank. Little influence of the initial temperature on the cavitation was observed. The temperature increment reached its peak when the flow rate was approximately $1.6\,\mathrm{m^3\,h^{-1}}$. The temperature increment decreased with the increase in fluid volume. Much more time was needed to heat the xanthan gum aqueous solution with an increased xanthan gum concentration.

Water and xanthan gum solution have been used in this study, but their heat conductivities are limited compared with conduction oil. For future field applications, thermal performance with conduction oil should be examined to improve the overall performance of gas hydrate prevention. Hydraulic cavitation can be realized in several ways, among which is the rotating propeller method used in this work. Downhole equipment corrosion may be a possible risk and needs special attention. Other methods, such as the methods used by Hydro Dynamic, Inc. and Cavitation Energy Systems, LLC, should be thoroughly evaluated with a numerical or experimental study. Besides the cavitation method in this paper, thermite reaction and tubing coating can provide us with some new thoughts in preventing and removing natural gas hydrate.

Data accessibility. Data available from the Dryad Digital Repository: https://dx.doi.org/10.5061/dryad.8gtht76nm [56].
Authors' contributions. M.W.: conceptualization, methodology, software, writing—original draft preparation, supervision. J.Q.: data curation, visualization, investigation. W.C.: writing—reviewing and editing.
Competing interests. We declare we have no competing interests.
Funding. This work was financially supported by the China Scholarship Council (grant no. 201806455009).

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
