## [Peer Review File · Royal Society Open Science]

Review History

RSOS-202054.R0 (Original submission)

Review form: Reviewer 1

Is the manuscript scientifically sound in its present form?

Yes

Are the interpretations and conclusions justified by the results?

Yes

Is the language acceptable?

Yes

Do you have any ethical concerns with this paper?

No

Have you any concerns about statistical analyses in this paper?

Yes

Recommendation?

Accept with minor revision (please list in comments)

Comments to the Author(s)

Much work has been done on the prevention of natural gas hydrate in chemical methods, yet little has been done from a physical way. The authors presented a novel way of handling the issue of natural gas hydrate and this approach may draw the attention from the petroleum industry. The subject is interesting, both numerical and experimental studies have been performed in the manuscript with necessary explanations. Before acceptance for publication, I would recommend several minor changes in the manuscript.

1. Major difficulties, challenges and original achievements of hydrate prevention could be highlighted in the introduction section.
2. Some limitations of this study, suggested improvements and future direction of this work could be highlighted in the conclusion section.
3. Hydrate formation and deposition information should be emphasized in the introduction section.

Review form: Reviewer 2

Is the manuscript scientifically sound in its present form?

Yes

Are the interpretations and conclusions justified by the results?

Yes

Is the language acceptable?

Yes

Do you have any ethical concerns with this paper?

No

Have you any concerns about statistical analyses in this paper?

No

Recommendation?

Accept with minor revision (please list in comments)

Comments to the Author(s)

- literature review part should be improved with recent gas hydrate production trials.
- the quality of figures should be improved

Decision letter (RSOS-202054.R0)

Dear Dr Wang

On behalf of the Editors, we are pleased to inform you that your Manuscript RSOS-202054 "Toward the development of cavitation technology for gas hydrate prevention" has been accepted for publication in Royal Society Open Science subject to minor revision in accordance with the referees' reports. Please find the referees' comments along with any feedback from the Editors below my signature.

Please submit your revised manuscript and required files (see below) no later than 7 days from today's (ie 05-Jul-2021) date. Note: the ScholarOne system will 'lock' if submission of the revision is attempted 7 or more days after the deadline. If you do not think you will be able to meet this deadline please contact the editorial office immediately.

on behalf of Professor R. Kerry Rowe (Subject Editor)
openscience@royalsociety.org

Reviewer comments to Author:

Reviewer: 1

Comments to the Author(s)

Much work has been done on the prevention of natural gas hydrate in chemical methods, yet little has been done from a physical way. The authors presented a novel way of handling the issue of natural gas hydrate and this approach may draw the attention from the petroleum industry. The subject is interesting, both numerical and experimental studies have been performed in the manuscript with necessary explanations. Before acceptance for publication, I would recommend several minor changes in the manuscript.

1. Major difficulties, challenges and original achievements of hydrate prevention could be highlighted in the introduction section.
2. Some limitations of this study, suggested improvements and future direction of this work could be highlighted in the conclusion section.
3. Hydrate formation and deposition information should be emphasized in the introduction section.

Reviewer: 2

Comments to the Author(s)

- literature review part should be improved with recent gas hydrate production trials.
- the quality of figures should be improved

===PREPARING YOUR MANUSCRIPT===

===PREPARING YOUR REVISION IN SCHOLARONE===

Author's Response to Decision Letter for (RSOS-202054.R0)

See Appendix A.

Decision letter (RSOS-202054.R1)

Dear Dr Wang,

I am pleased to inform you that your manuscript entitled "Toward the development of cavitation technology for gas hydrate prevention" is now accepted for publication in Royal Society Open Science.

on behalf of Prof R. Kerry Rowe (Subject Editor)
openscience@royalsociety.org

Response to the reviewers

First of all, on behalf of all the authors, I'd like to show my sincere gratitude to the reviewers and editors. Your expertise, patience, and valuable time have greatly enhanced the quality of the manuscript. The following is the response to the reviewers' comments.

Reviewer 1:

Much work has been done on the prevention of natural gas hydrate in chemical methods, yet little has been done from a physical way. The authors presented a novel way of handling the issue of natural gas hydrate and this approach may draw attention from the petroleum industry. The subject is interesting, both numerical and experimental studies have been performed in the manuscript with necessary explanations. Before acceptance for publication, I would recommend several minor changes in the manuscript.

1. Major difficulties, challenges and original achievements of hydrate prevention could be highlighted in the introduction section.

The introduction part has been revised according to the reviewer's comment. In the manuscript, words with blue color are revised one and words with black color come from the original manuscript. Hydrate formation can happen at places where high-pressure and low-temperature exist. It's tricky to handle hydrate blockage and the risks are high. The Deep Water Horizon explosion has brought 11 lives to death and we can't afford any hydrate blockage accident with the loss of human beings. The key to hydrate treatment is to eliminate any possibility that can cause hydrate formation. Take precautions before they happen. The method proposed in this paper is an active defense. For gas production or transportation, we can raise the temperature around the tubing or

pipeline any time just to prevent the formation of gas hydrate. The cavitating heater can be installed on the seabed and controlled at the surface.

2. Some limitations of this study, suggested improvements and future direction of this work could be highlighted in the conclusion section.

For the limitations of this study, experimental observations, former research have both demonstrated the feasibility of the cavitating heating method. One big problem for such a method is its installation risk. For natural gas well, the umbilical pipe needs to be reeled around tubing and lowered into the annulus. Once the umbilical pipe is damaged during the installation, the whole pipe configuration has to be replaced. A possible solution for this is to integrate the heating system with tubing or cover the umbilical pipe with hard-facing materials. The cavitator is installed on the seabed and communicates with engineers through cables, data transmission for several thousand meters may also be a possible problem. The conclusion part has been revised addressing the above questions.

3. Hydrate formation and deposition information should be emphasized in the introduction section.

In the revised version of the manuscript, much more words have been spent on explaining the formation and deposition of water hydrate. About 20 published papers or conference proceedings have been added to the reference list just to emphasize this part. Only after the reader form a basic understanding of how gas hydrate is formed and deposited can they develop a relatively deep understanding of the method proposed here.

Reviewer 2:

1. literature review part should be improved with recent gas hydrate production trials.

The literature review part has been revised and improved with gas hydrate formation

and deposition, physical and chemical methods with their advantages and disadvantages. Hydrate crystals trap and temporarily store large quantities of methane gas molecules. When the thermodynamic conditions change, the hydrate crystals then are decomposed and these gas molecules are released, bringing us considerable natural gas energy. Many of the above hydrate control methods can be used directly without modification for the efficient development of hydrate resources, such as depressurization and thermal injections. The United States, Canada, Germany, Russia, Japan, India and other countries have all carried out hydrate-related development work. Specifically, the United States has conducted hydrate test trials in Alaska, Japan in Nankai and China in the Shenhu of the South China Sea with various development achievements.

2. the quality of figures should be improved

All figures and tables used in the manuscript have been double-checked and regenerated for improved qualities.